# Functional MRI in Radiology—A Personal Review

**DOI:** 10.3390/healthcare10091646

**Published:** 2022-08-29

**Authors:** Martin Lotze, Martin Domin, Sönke Langner, Thomas Platz

**Affiliations:** 1Functional Imaging Unit, Institute of Diagnostic Radiology and Neuroradiology, University Medicine Greifswald, 17475 Greifswald, Germany; 2BDH-Klinik Greifswald, Institute for Neurorehabilitation and Evidence-Based Practice, “An-Institut”, University of Greifswald, 17489 Greifswald, Germany; 3Neurorehabilitation Research Group, University Medical Centre, 17489 Greifswald, Germany; 4AICURA Medical GmbH, 12103 Berlin, Germany

**Keywords:** functional magnetic resonance imaging, structural MRI, history of fMRI, radiology in Germany

## Abstract

We, here, provide a personal review article on the development of a functional MRI in the radiology departments of two German university medicine units. Although the international community for human brain mapping has met since 1995, the researchers fascinated by human brain function are still young and innovative. However, the impact of functional magnetic resonance imaging (fMRI) on prognosis and treatment decisions is restricted, even though standardized methods have been developed. The tradeoff between the groundbreaking studies on brain function and the attempt to provide reliable biomarkers for clinical decisions is large. By describing some historical developments in the field of fMRI, from a personal view, the rise of this method in clinical neuroscience during the last 25 years might be understandable. We aim to provide some background for (a) the historical developments of fMRI, (b) the establishment of two research units for fMRI in the departments of radiology in Germany, and (c) a description of some contributions within the selected fields of systems neuroscience, clinical neurology, and behavioral psychology.

## 1. Introduction

The history of the rise of fMRI and the most important researchers initiating its usage were overviewed in a special edition entitled “20 years of fMRI” in *Neuroimage* about a decade ago [1]. International annual meetings of the human brain mapping community have existed since 1995.

We, here, do not provide an overview of the 30 years of fMRI; however, we offer a very personal view on the beginning of fMRI in Germany and our experiences in the 16 years of an fMRI research unit in one of the smallest medical faculties in Germany—the University Medicine of Greifswald. We do so because running an fMRI group in a university radiology department is a challenge not often practiced, and it might be a valuable strategy to integrate methodological and neuroradiologic know-how with interests in neuroscientific research. Historically, but also practically, most of the research units dedicated to fMRI are located in neurological and psychiatric institutes, although radiology and neuroradiology have an important impact on the development of a number of methods, especially at its beginning [2,3,4].

The different fields of research overviewed here are dependent on several factors which interact with each other: (a) the passion and interest of the individual researcher, (b) the possibilities offered by a research unit with respect to MRI equipment, equipment for technical issues needed for fMRI, and technical support, (c) local and international cooperation partners, who provide access to additional methods or patients, and (d) the funding resources available.

## 2. Methods and Material 

### Development of the fMRI Method

In 1990, Ogawa observed the blood oxygenation level-dependent (BOLD) effect in T2*-weighted MRIs of rodents [5]. This was based on the reports by Keith Thulborn in 1982, who described the changes in magnetic properties of hemoglobin by oxygenation [6], and Linus Pauling’s finding in 1935 that the hemoglobin molecule changes in its dependence on oxygenation. In 1991, at the tenth annual meeting of the Society for Magnetic Resonance in Medicine in San Francisco, Belliveau demonstrated the first functional imaging results in man by using contrast media. The advent of contrast-medium-free functional imaging has been described in detail by R. Turner [7]: “the first MRI observations of BOLD functional activation in humans were made in 1991 within a few months of each other, independently, at the MHG-NMR Center, and the MRI lab at the University of Minnesota”. Overall, several researchers contributed to the invention of fMRI, and the paper, which showed (for the first time) convincing data of an approximate 2% BOLD effect by using echo-planar imaging (EPI) in the visual cortex by flash-light stimulation, was published by Kwong, Turner, and others in 1992 [8].

The potential of fMRI was immediately recognized since, at that time, most researchers interested in functional imaging applied PET imaging, which showed a considerable radiation burden and therefore did not permit longitudinal observations [9]. Further, in 1992, Frahm and colleagues [4] published the first results using the fast low-angle-shot (FLASH) MR technique during photopic stimulation of the visual cortex and described the onset, rise, and duration of the BOLD effect. In contrast to FLASH, the EPI technique offered the possibility of imaging more than five slices but showed a considerable susceptibility to artifacts and distortions [10]. However, this technique was initially possible only in a low number of MRI-scanner prototypes.

Several other young scientists from Germany contributed to the early rise in functional imaging experiments and its embedment in neuroscience: some of them did their research training at the University College in London (UCL); a group headed by K. Friston, R. Turner, and R. Frackowiak. These three researchers played an especially important role in spreading fMRI in Europe and, together with others, they developed the statistical parametric mapping (SPM) software, first to be applied to the evaluation of PET data only, but later also to fMRI [11]. Other researchers from Germany, predominantly those interested in the visual system, joined the Harvard-based research group (with, among others, P.A. Bandettini and K.K. Kwong). The method of fMRI was fast developed with respect to the optimization of spin-echo imaging sequences [12], increasing the BOLD magnitude and decreasing noise [13], optimizing spatial and temporal resolutions [14], and investigating the temporal characteristics of BOLD [2]. It was also important to discover how the BOLD effect developed during simultaneous direct neural recordings and experiments performed on the awake monkey [15,16].

## 3. Results

### 3.1. fMRI in Tübingen (Personal View from 1996–2006)

Tübingen was one of the first university radiology departments in Germany, headed by a C4 professor for neuroradiology; K. Voigt established Neuroradiology as an independent department from Diagnostic Radiology (headed by C.D. Claussen, who came from the FU Berlin to Tübingen in 1988). W. Grodd, as a neuroradiologist, was especially interested in establishing an fMRI group, and after receiving an invitation for a professorship for neuroradiology at the FU Berlin in 1994, he was asked to construct a section of experimental nuclear magnetic resonance imaging of the CNS in Tübingen in 1995. This section was equipped with sufficient manpower for establishing the first fMRI center in German radiology. The Neuroradiology Tübingen took part in the Human Brain Mapping World Organization meetings right at the beginning, in Paris in 1995 and in Boston in 1996. Since then, the members of that section continuously contributed to these annual world meetings—later (1997 in Copenhagen) entitled the Organization for Human Brain Mapping (OHBM). From the very beginning, it integrated physicians (predominantly radiologists, neurologists, and psychiatrists), physicists (the best-known contribution for fMRI is by M. Erb), and computer scientists and psychologists (e.g., the N. Birbaumer fMRI lab). Over the initial years, the group developed its scope of research from initial core topics, such as language and speech representation (H. Ackermann, D. Wildgruber [17]), motor representation (W. Grodd, and since 1996, also M. Lotze [18]), psychology (headed by N. Birbaumer and H. Flor [19]), psychiatry (memory: R. Heun and F. Jessen from Bonn [20]), anxiety and emotional processing (R. Schneider and U. Habel, Aachen [21]), the feeling of self (T. Kircher), and pediatric research (W. Grodt, I. Krägeloh-Mann, and since 1999, M. Staudt [22]).

During the second part of the 90 years of the last century, the science facilities in Tübingen rose to be a major player in German neuroscience with high-ranking neuroscience groups at the university, the University Medicine, and the Max Planck Institutes, with a very successful gathering of neuroscience funding since then. The high amount of research funding enabled researchers—and also physicians—to work longer periods completely in fMRI research [23], a basis for the continuous development of methods and knowledge bases.

Researchers continuously developed new data evaluation strategies at the beginning of the fMRI evaluation. For instance, spatial realignment strategies were completely absent in the beginning, and spatial normalization procedures were far from today’s quality. Advanced normalization procedures were developed (e.g., for the human cerebellum), enabling a demonstration of a somatotopic cerebellar sensorimotor representation [3] in a spatial resolution, which was only possible in an invasive manner before. These development activities and the critical testing of tools not only nurtured an individual responsibility culture in each researcher for her/his data, but also helped to spread methodological knowledge across the neuroscience community.

Early on, the group integrated other neurophysiology research methods (MEG, EEG, TMS) and psychophysiological measures into their fMRI mapping results. They placed a major emphasis on controlling for both: (1) documentation of the actual presentation in the fMRI experiment, and (2) the performance or interaction of the participant investigated. The latter comprised the control of the motor response [24], psychophysiology [25], and eye-tracker-based startle responses [26]. These methods could then be used for investigating the central mechanisms of emotional [27] and social interactive [28] processing or the understanding of humor processing in the brain [29]. In addition, the high innovative power of scientific collaboration enabled extremely unconventional approaches in conditional learning [30], brain modulation [31], aggression research [32], fMRI-navigated functional lesioning [33], and training using fMRI feedback [34]. In addition, a number of projects investigated research fields that overlapped with music production, such as one investigating brain processing while playing an instrument in professional violinists [35] and also on singing performance [36,37]. Both interests in new topics and realistic work on the artifacts induced by singing in a lying position in an fMRI scanner were the basis of this work. On the presentation and stimulation side, M. Erb and others (C. Braun from the Center of Magnetoencephalography) established highly elaborate somatosensory (air puff and electrical stimulation), pain (thermal, tactile, and electrical), visual (predominantly via beamer and video-splitter system presentation environments), and auditory (stimulation and recordings) stimulation paradigms for the fMRI. Equipment tools developed then have been continuously in use [38] (see Figure 1 and Figure 2).

### 3.2. Building up Research and Funding in the Greifswald fMRI Unit

Alfons Hamm, a biological psychologist who studies the human processing of emotions, was especially interested in detecting the neural processing pathways for the fear response. Mapping these pathways is quite challenging since the limbic system is highly prone to artifacts (e.g., the air in susceptibility artifacts due to the location near the frontal sinus). Therefore, the experiments which investigate and provoke a fear response in the MRI are methodologically challenging. The Tübingen group around N. Birbaumer started very early by quantifying the BOLD magnitude in the amygdala in response to aversive stimuli and also included patient groups, such as those with social phobia, in their protocols [39]. Later developments using fMRI feedback [40,41] profited from the methodological knowledge of these early years. It was, therefore, of high interest for biological psychologists, such as A. Hamm, to establish a local cooperation partner who already had experience with fMRI of emotional processing.

N. Hosten, who received a professorship for Diagnostic Radiology and Neuroradiology at the UMG in Greifswald in 2001, was open to the idea of establishing a Functional Imaging Unit in his center. After several additional years, this made it possible for the functional imaging group in Greifswald to be officially established in 2006. In addition, a new 3T-MRI system was installed, which provided considerable new possibilities (e.g., measurement time of two workdays per week with the option to measure on weekends, offering the possibility to measure with external partners for four days in a row).

The group was initially based around the neurologist and neuroscientist M. Lotze, the computer scientist, M. Domin, who was primarily interested in structural and diffusion imaging methods, and two physicists (2006–2011: E. Kaza; 2011–2018: J. Pfannmöller). The team was highly motivated in setting up the methodological equipment, and the main topics they focused on were: (1) emotional processing and anxiety, (2) sensorimotor plasticity research, including stroke research, and (3) chronic pain.

Integration of the Unit into the “Verbund Neuroimage Nord”, which comprised the MRI units of three Northern German Universities (Hamburg (C. Büchel), Lübeck (F. Binkofsky), and Kiel (H. Siebner)) were not successful, but there was the possibility for an association with the Bernstein Center Berlin (JD. Haynes). Later cooperation regarding work on nonlinear fMRI-evaluation algorithms was related to that association [42]. Overall, the small size of the imaging group made it difficult to realize the interactions with methodologically highly advanced groups. However, in some projects—without external funding for the Greifswald group—this could be partially realized [43].

Since the University of Greifswald did not focus on neuroscience, such as Tübingen, starter-funding was based on in-house support. Funding from the German Research Foundation (DFG), which remained the major funding resource for the unit, was first achieved for the research on motor plasticity following a stroke. This was achieved in cooperation with T. Platz, who was appointed as head of the Department for Neurorehabilitation in Greifswald in 2006.

### 3.3. Neurorehabilitation Studies

T. Platz, from the neurorehabilitation department of the Klinik Berlin, in association with the chair of neurorehabilitation at the FU Berlin (K.-H. Mauritz), Germany, was offered the position as the new director of clinical research of the Greifswald BDH neurorehabilitation hospital. The BDH clinic Greifswald had an extraordinary position in German neurorehabilitation because it was established in 1998 as a purpose-built modern unit, which was also used for severely impaired head trauma and spinal cord injured patients and had comprehensive care, ranging from intensive care to day-care clinics, and it had the status of a university institute academically (“An-Institut”). Together with the high quality of neurorehabilitation, the research interests of T. Platz and the location near the clinical campus (and the MRI-investigation unit), the unit was unique in longitudinal intervention studies in neurorehabilitation. An initial study on upper limb motor function in chronic stroke patients was published in 2012, describing the spectrum of biomarker measurements and the most promising correlates for recovered function [44]. During his time in Berlin, T. Platz developed an effective strategy for upper limb rehabilitation training [45,46] (the “impairment-oriented training”, IOT), which was highly suited for investigating longitudinal intervention studies, obtaining behavioral progress and biomarkers following motor impairment after brain damage, such as a stroke. Alongside the clinical and behavioral data (sensorimotor testing), other MRI biomarkers (lesion mapping and white matter damage quantification), fMRI biomarkers (the recruitment of motor areas during successful recovery) and TMS measures (the excitability of the motor cortex, modulation of cortical excitability during training) were integrated into the intervention studies applying IOT. In addition, IOT (and here, we used arm ability training [47] (AAT)) could also be applied in healthy participants to increase their upper limb performance to a considerable extent [48]. It could be further used to investigate the mechanisms of cortical excitability changes due to the training of the non-dominant upper limb [49] and for investigating possible cortical modulation of motor training by theta-burst TMS [50]. It was also successfully combined with somatosensory priming of the fingertips, which even increased the performance in hand-grip force overtraining [51]. DFG funding was granted for a longitudinal intervention study, which demonstrated that the ventral premotor cortex of the affected hemisphere contributed to recovered upper limb performance in the subacute stage following a stroke [52]. Especially for the acute stage following a stroke, we determined the prognostic role for a stepwise regression of optimized biomarkers for predicting upper limb motor outcomes [53] after 3 or 6 months. The biomarkers for recovery potential and the stratification of evidence-based treatment are of high importance in a rehabilitation system with increasing numbers of patients and decreasing personal resources [54]. However, research on the swallowing function after post-stroke dysphagia is rarely performed. The DFG funded our research on the representation of swallowing and recovered the swallowing function in post-stroke dysphagia. The method for that was based on the developments to control for artifacts and problems associated with the lying position for investigating the swallowing function, performed 10 years before with M. Erb in Tübingen. Our research enabled us to describe the swallowing network, including brain stem representation [55] and the sequential representation of swallowing using ultrafast fMRI [56]. The aim was to investigate patients with recovered dysphagia by applying a swallowing task in the fMRI (see Figure 2). Here, the essential role of a recruitment of primary somatosensory (S1)-somatotopic representation sites during recovered swallowing performance was shown [57]. Later, the important role of callosal white matter integrity for a bihemipsheric S1-interaction could be demonstrated for the same clinical dataset [58]. For all these studies, the age-matched controls had to be investigated, resulting in publications on age-related functional representation changes for hand motor function [59] and swallowing function [60].

### 3.4. Cooperation Studies with Local Medical Partners

In addition to the studies on stroke and pain, research was conducted with two major cooperation partners, which did not obtain funding but were nevertheless successful in hitting the top ten list of best-cited studies from the fMRI unit (Table 1).

In cooperation with Neuropediatrics (H. Lauffer), children with a high risk for obesity were investigated with regard to their functional representation of emotionally relevant pictures, such as pictures of food and sports. These activation maps were compared to children with normal age-related weight [61]. Interestingly, children with a high risk for obesity showed a decreased response to positively valenced pictures of all thematic fields (social, emotional, food and sports), indicating lower emotional variability in comparison to children with no risk for obesity. That study received the prophylaxis award from the German Society for Obesity in 2013. The second cooperation study was realized with the Department of Neurology, Greifswald, and the University of Hildesheim and focused on the research on verbal creativity. We, here, applied a creative writing real-life fMRI design to investigate medical students [66] and students studying creative writing [71]. This graduation work received an award as the best graduation work of the University and the results evoked considerable international press attention (see https://www.nytimes.com/2014/06/19/science/researching-the-brain-of-writers.html?action=click&contentCollection=Science%AEion=Footer&module=MoreInSection&pgtype=article, accessed on 18 August 2022).

The research methods used in functional imaging and research in diagnostic radiology differ substantially. One study, which integrated the knowledge of a group data evaluation used in fMRI with clinical research, was a study on gadolinium-based contrast agents for diagnostic purposes in multiple sclerosis patients [72]. After the initial report [73] on the increased signal intensity of the dentate nucleus following serial injections of linear gadolinium-based contrast agents, a multitude of studies have since been published. This signal increase is a surrogate marker for gadolinium deposits in brain structures. However, in all these studies, the region of interest to be evaluated was chosen a priori and manually segmented. To overcome these limitations, S. Langner and M. Domin applied the image preprocessing pipelines normally used in fMRI studies. Using the “Diffeomorphic Anatomical Registration using Exponentiated Lie Algebra” (DARTEL) normalization process, they performed a voxel-based whole brain analysis in patients who underwent repeated administration of macrocyclic gadolinium-based contrast agents. This was the first study that was able to demonstrate, in a voxel-based analysis, that macrocyclic contrast agents do lead to a signal increase on plain T1-weighted images.

### 3.5. Cooperation with Psychology on fMRI in Emotional Processing

From the beginning, the methodological interaction with the group belonging to A. Hamm was highly intense, and especially with J. Wendt, a number of important contributions to psychophysiology and the understanding of classical conditioning and the fear network could be realized [63,69,74] (see Table 1). Especially for these network studies, standardized emotional fMRI-paradigms had to be established over many years in constant quality, first in crossover investigations and later in longitudinal intervention studies on anxiety patients [75]. There was only one common funding between the fMRI unit and psychology, and that was a study supported by the DFG on the impairment in emotional processing in stroke survivors with insular damage. Although this study required an immense effort (patients’ selection and testing; psychophysiology and MRI measures), most of the results could be published years after the funding period [76]. 

### 3.6. Pain Research

An important part of the work and cooperation in anesthesiology (T. Usischenko) and dentistry (B. Kordass) was the investigation of chronic pain. These comprised the mechanisms of pain modulation by trans-auricular vagus stimulation [77]), which received DFG funding later on. Further, B. Kordass was interested in the biomarkers for temporomandibular disorder (TMD) and adaptation of the brain to interventions, and later the DFG funding was focused on the representation of occlusal movements and chewing [55], the longitudinal effect of maxillar [78] and mandibular [79] splints on occlusal movements in TMD. Another direction was the investigation of neuropathic pain, which had already been very successfully investigated together with H. Flor in Tübingen beforehand [80]. Together with our cooperation partner, G.L. Moseley from Sydney and Adelaide, Australia, we now intend to focus on fMRI, MRI, TMS, and behavioral testing biomarkers in complex regional pain syndrome (CRPS). Although not funded by the DFG, we later achieved funding from the “Else Kröner-Fresenius-Stiftung”, which enabled us to perform a larger intervention study that investigated biomarkers for the graded motor imagery (GMI: a behavioral intervention, increasingly inducing movement in the painful limb) treatment-associated effects in chronic CRPS [38]. Overall, fMRI biomarkers have a high potential in planning and supervising interventional studies on chronic pain, and will be most challenging for functional imaging in the coming years.

Two additional funding grants for bilateral cooperation were associated with the abovementioned project: firstly, BMBF-funded cooperation with W. Byblow in Auckland, New Zealand, for the exchange in scientists and investigations on the mechanisms of treatment options on cortical excitability in chronic pain was achieved [81]. Secondly, binational funding was achieved for mental training, together with F. Lebon from the Department for Sports Science in Dijon, France—a colleague from an experiment in Auckland on cortical processing during imagery and mental rotation. This project enabled new investigations on the cerebral representation of mental training [82,83].

### 3.7. Methodological Developments

The fMRI unit Greifswald is especially known for its longitudinal fMRI monitored studies in the field of neurorehabilitation and chronic pain. This covers upper limb function impairment following a stroke [84], GMI in CRPS [85], and splint-based interventions in temporomandibular disorders [78] (TMD).

For all these fields, special solutions were developed to enable performance control and balance in the fMRI scanner over time. For the TMD-intervention studies, different types of occlusal splints were tested, and treatment effects were associated with functional performance and imaging biomarkers [79]. In addition, the DFG funding was based on the development of an occlusal pressure system monitoring performance in the MRI (see Figure 2) in intervention studies and on the investigation of psychological factors (stressors, anxiety, extinction learning). Another research direction was the continuous work on the quantification of white matter integrity following a stroke and multiple sclerosis, based on the experience of M. Domin in these methods. If specific white matter tract masks had not been described before, specific new masks were obtained on the basis of the DWI measurements from the Human Connectome datasets (HCP) [86], and these could then be applied to patient data [58]. This method enabled new knowledge about more complex emotional processing interactions in patients with stroke-caused lesions of the insula—a DFG study challenging the interaction of stroke researchers and psychologists.

For the evaluation of the somatotopic representation of the human fingertips and the possible distortion of the somatotopic ordering in CRPS, the limited spatial resolution of our 3T Verio was a large challenge. J. Pfannmöller, who later joined the Polimeni group at the Martinos Center at Harvard, Boston, developed measuring methods and data evaluation tools, which enabled an increase in the spatial resolution, up to about 1.5 mm, which was enough to demonstrate the mechanisms on a 3 Tesla system, known to be valid for earlier invasive animal research. This method reliably demonstrated a decrease in the primary somatosensory cortex finger representation fields in CRPS patients [38,87,88,89] and is an excellent example of the advances that can be achieved by years of careful experiments to extend the limits of functional imaging, even with a 3 Tesla MRI.

### 3.8. Funding Landscape and National and International Cooperation

After considerable “start” research funding by the faculty, a later shut down of the neuroscience research interest group of the medical faculty hampered further intramural funding. In addition, performance-related money, which is associated with a performance in achieving competitive national funding or high-ranking publications, was almost absent. On the other hand, the opportunity to work in foreign research groups was an important booster for international cooperation and publishing, especially for pain research [85,90,91].

Most of the cooperation with other German and Swiss Universities was already laid out in the decade before in Tübingen. This comprised of S. Anders, Lübeck (social neuroscience), Elise Wattendorf, Fribourg (emotional processing of ticklish laughter), Dirk Wildgruber, Tübingen (emotional processing of facial expressions and sounds), Karen Zentgraf and Jörn Münzert, Frankfurt/Giessen (mental training and representation), and Simon Eickhoff, Jülich (innovative fMRI evaluation methods and meta-analyses). Besides the panic network participation of the psychologists, we never had the chance to participate in national multicenter studies—probably because of a lack of patient access in a scarcely populated rural area. Last but not least, the location of Greifswald, more than three hours away from the next international airport, was an additional issue for our flexible interaction studies.

## 4. Discussion

### 4.1. A Trend from Functional to Structural Big Data Evaluation

The University of Greifswald has a history of longitudinal epidemiologic cohorts with whole-body MRI included. Although these cohorts offer the opportunity for an evaluation of thousands of brain images and are longitudinal in their conceptualization, an MRI of the brain is predominantly restricted to a T1-weighted structural image (magnetization-prepared rapid gradient-echo imaging (MPRage)) and most of the questionnaires for cognitive function, motor performance testing, or the questionnaires on pain experience are not adjusted to international standards. The results obtained with these data are restricted due to historical reasons (consistent scanner hardware and software over time) and limited imaging quality (in comparison to other international cohorts). We performed a number of cross-sectional investigations on grey matter volume in association with lifestyle factors (e.g., nicotine consumption [62], income and education [92], gender [68], and leisure sports [93]) in thousands of participants. The unique challenge, however, was a longitudinal evaluation of the epidemiologic datasets [94], which is currently funded by the German Research Community (DFG LO 795/37-1).

### 4.2. Functional MRI in Radiology—Pros and Cons

Overall, a functional MRI is embedded in a number of structural analysis techniques used in MRI, and the techniques first developed for fMRI-data evaluations can also be applied (at least in a modified technique) in structural MRI evaluations, such as in large datasets. For the unique characterization of activation fMRI, the control of presentation and behavior inside the scanner is crucial. Therefore, an integration of psycho- and neurophysiology with functional imaging is obvious. In addition, the low temporal resolution of fMRI calls for the integration of methods investigating the time course of activation in more detail. Since fMRI is a technique based on changes in voxel-based intensity values over time (which expresses changes in the BOLD effect), it cannot be evaluated in a way neuroradiologists are used to analyze clinical data. Voxel-wise effects are compared with a hemodynamic response curve. There are a number of physiological and statistical assumptions, which are continuously refined, but have been standardized with respect to processes and batch processing in fMRI-data evaluations (for an overview, see [95]). It is essential that each of these processes is performed with high methodological care and knowledge, and quality control is of utmost importance. In addition, any changes to the default procedures of batch processing of fMRI evaluation have to be performed in a deliberated way, which has to be described and justified in the Methods section. Especially when considering the number of different evaluation strategies and the limited reproducibility, well-defined and optimized procedures are mandatory. However, different hypotheses often call for individual testing strategies, which limit the attempts for standardization. The latest quality standards are, therefore, asking for a clear evaluation strategy for observational studies, too, which are recorded in a predefined trial. Overall, neuroradiologists are extremely well-trained in neuroanatomy and recognizing pathology in images. This helps, not only in quality checking and the identification of damaged brain tissue, but also in the neuroanatomical localization and interpretation of functional data.

### 4.3. Limitations

fMRI, with the need for active participation by the patient, has gained only limited practical use in clinical testing (predominantly for motor and language tasks; for an early overview, see [95]). Especially with stroke rehabilitation of motor function, the role of fMRI was disappointing, and in the recent algorithms predicting upper limb outcome following stroke, even diffusion imaging, quantifying pyramidal tract integrity, is a relevant additional predictor only in a small subset of patients [96]. More automatized evaluation strategies and the selection of more holistic parameters might provide a step forward for using fMRI as a valid biomarker.

### 4.4. Future Directions

Furthermore, topics not often investigated in recovery studies following brain damage are highly promising for future research. Here, stratification for optimized treatment might be performed with neuroimaging biomarkers. One of these fields is neurogenic dysphagia. With respect to the swallowing function after a stroke, the latest reports on the integrity of specific tracts for later swallowing recovery might be an important biomarker [58]. Task-free fMRI, which is much easier to measure, but is challenging with respect to evaluation strategies [97], showed low reproducibility between clinical samples [98] or low predictive power as a biomarker [53]. However, for instance, for the proper location of repetitive TMS brain stimulation in depression, navigation with resting-state fMRI has shown promising results lately [99]. In addition, the assessments of more global parameters, such as a degree rank-order in a graph theoretic approach, might well provide a more robust differentiation, at least in sorting out patients with chronic pain and healthy participants [100,101]. Especially in the field of chronic pain, the fMRI biomarkers of activation studies are gaining increasing importance [38,64,85,102]. Furthermore, the measurement of the temporal evolution of structural connectivity seems to have explanatory value for functional recovery that could otherwise not be explained [103].

The role of emotional processing deficits is another field that is widely unexplored for the patient group with brain damage [104]. Highly specialized centers might therefore provide these methods for clinical use in the future.

There is a remaining high interest among researchers in the usage of these methods in basic neuroscience, and the community for OHBM is young and integrates many new possibilities and tools. Especially, the possibility to integrate datasets from different centers, for instance, by the ENIGMA interest groups [105], can overcome discrepant results from different samples due to a lack of power, particularly for patient studies [106] and for developing evaluation strategies in large datasets [107]. However, funding organizations have to overcome the tradition of predominantly providing funding for new investigations rather than for more advanced evaluation procedures of larger, already existing datasets. Although the scientific community is increasingly trained with advanced computer science methods, the capacities in experimental psychology, neuroanatomy, and neurophysiology seemed to decrease over the last decade. This makes it even more necessary to invest more resources into integrated training in neuroscience. Neuroradiology can provide an important role, especially for these educational needs, since people working there are used to integrating knowledge about MRI imaging techniques, data quality, practical issues on patient investigation and care as well as experiencing limitations on the interpretation of the datasets.

## Figures and Tables

**Figure 1 healthcare-10-01646-f001:**
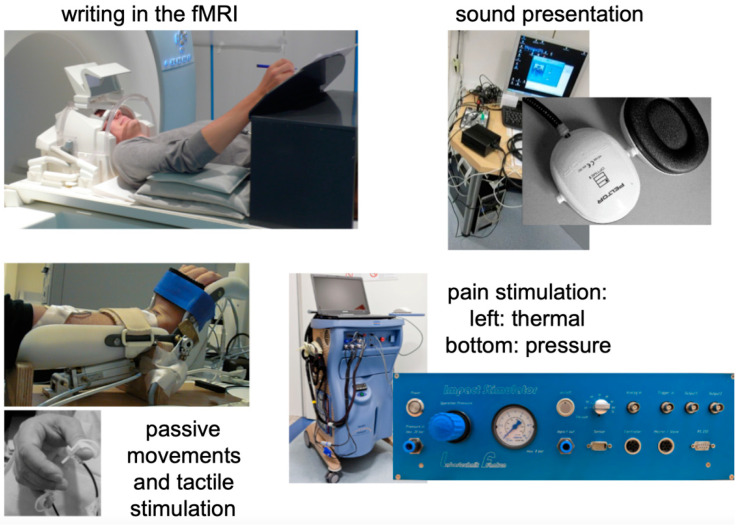
Experimental settings for the presentation of stimuli in the functional MRI scanner. **Left top**: writing table for fMRI investigations with a doubled-mirror system affixed above the head coil; **right top**: sound presentation system; **left bottom**: pneumatic stimulation system; **right bottom**: pain stimulation devices for thermal and pressure pain stimulation.

**Figure 2 healthcare-10-01646-f002:**
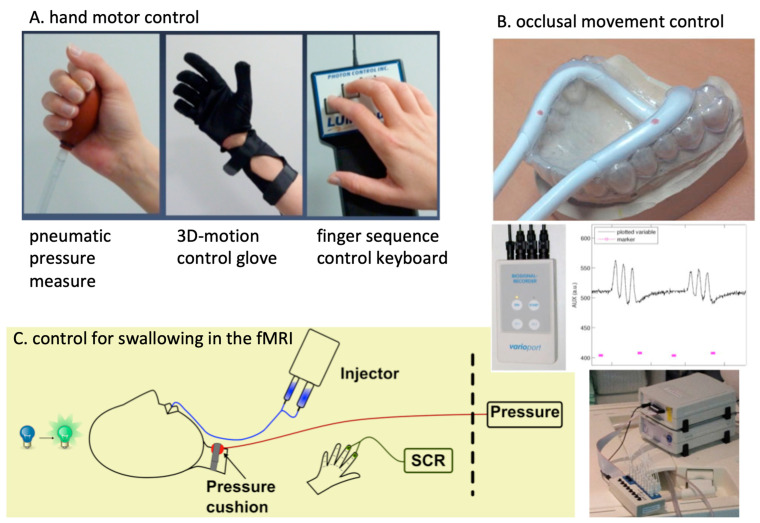
Control of motor performance during fMRI tasks. **Left top**: three possibilities to control hand motor activity: pneumatic ball, coupled with a pressure measurement device; glove, commercially distributed for virtual reality environments and suitable for MRI; finger sequence control keyboard. **Right top**: development with an individual splint for controlling occlusal pressure and frequency (see plot **middle right**) with the Varioport device (Varioport; Stuttgart; **middle left**). **Bottom**: control of swallowing by a pressure cushion combined with the Varioport device. Skin conductance response (SCR) can be measured with Brain Vision Analyzer 2.0 (Brain Products, Gilching, Germany; **bottom right**).

**Table 1 healthcare-10-01646-t001:** Papers from the fMRI unit that were referenced more than 50 times.

Area of Research Referenced in (Web of Science) in 7/22	Partner	Topic	First Author, Year, Citation	Title
Cross-sectional fMRI study on children (112)	Pediatrics UMG	Obesity	Davids S, 2010, [61]	“Increased dorsolateral prefrontal cortex activation in obese children during observation of food stimuli”
Voxel-based morphometry from SHIP data (95)	Community Medicine UMG	Life Style factors, brain atrophy	Fritz HC, 2014 [62]	“Current Smoking and Reduced Gray Matter Volume-a Voxel-Based Morphometry Study”
fMRI on phobia-related stimuli (95)	Psychology University	Emotion processing	Wendt J, 2008 [63]	“Brain activation and defensive response mobilization during sustained exposure to phobia-related and other affective pictures in spider phobia”
Review (88)	External (Australia)	Biomarkers in CRPS	DiPietro F, 2013 [64]	“Primary Somatosensory Cortex Function in Complex Regional Pain Syndrome: A Systematic Review and Meta-Analysis”
ALE meta-analysis (83)	Jülich	Pain biomarkers	Friebel U, 2011 [65]	“Coordinate-based meta-analysis of experimentally induced and chronic persistent neuropathic pain”
Functional MRI on creative writing (83)	Neurology, University of Hildesheim	Verbal Creativity	Shah C, 2013 [66]	“Neural correlates of creative writing: An fMRI Study”
Voxel-based morphometry from SHIP data (73)	Community Medicine UMG	Chronic back pain; brain atrophy	Fritz HC, 2016 [67]	“Chronic Back Pain Is Associated With Decreased Prefrontal and Anterior Insular Gray Matter: Results From a Population-Based Cohort Study”
Voxel-based morphometry from SHIP data (70)	Community Medicine UMG	Gender identification, paper 1	Lotze M, 2019 [68]	“Novel findings from 2838 Adult Brains on Sex Differences in Gray Matter Brain Volume”
fMRI, MRI, and TMS on chronic stroke (61)	Neurology UMG, Auckland NZ	Biomarkers on upper limb function	Lotze M, 2016 [44]	“Contralesional Motor Cortex Activation Depends on Ipsilesional Corticospinal Tract Integrity in Well-Recovered Subcortical Stroke Patients”
fMRI on phobia-related stimuli (55)	Psychology University	Emotion processing	Holtz K, 2012 [69]	“Brain activation during anticipation of interoceptive threat”
fMRI on emotional processing (53)	Fribourg, Switzerland	Ticklish laughter	Wattendorf E, 2013 [70]	“Exploration of the Neural Correlates of Ticklish Laughter by Functional Magnetic Resonance Imaging”

## Data Availability

No supporting data have been are provided for that personal review.

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
