# Peer review of "Functional MRI in Radiology—A Personal Review"

_healthcare, 2022, doi:10.3390/healthcare10091646_

Round 1

Reviewer 1 Report

Dear authors,

Keywords

Please use mesh terms

Introduction

Is too short and has no references

Also, the aim is not provided

Material and methods section is not specific

Please provide personal methods used for this study

The paper is not well organized

Limitations and future directions are missing

What is the novelty of this manuscript?

References are not in accordance with the journal style

Author Response

Reviewer 1

Dear authors,

Keywords: Please use mesh terms

Response: we rechecked all keywords for being registered as mesh terms by https://meshb-prev.nlm.nih.gov/search. We now completely changed the key words to “Functional magnetic resonance imaging, structural MRI, history of fMRI, Radiology in Germany.”

Introduction: Is too short and has no references. Also, the aim is not provided.

Response: We now added some references in the Introduction. We apologize for our rather superficial adaptation of our review into the formal demands of the journal. We also added an aim in the Introduction now.

Material and methods section is not specific; Please provide personal methods used for this study.

Response: We inserted on page 3: “This is a personal review of the first author on developments in a field of structural and functional MRI. It is selectively chosen predominantly for the topics of systems neuroscience (and here for plasticity research and training), parts of clinical neurology (stroke, chronic pain), and behavioral psychology (especially emotional processing). We would like to mention some methodological developments first which were important for preparing the field.”

The paper is not well organized.

Response: We apologize for the adaptation to the guidelines of Healthcare which in fact are not ideal for personal reviews (for which we were invited).

Limitations and future directions are missing

Response: We now added limitations and future directions to the Discussion.

What is the novelty of this manuscript?

Response: The novelty is a personal review over the usage of MRI methods in the field of clinical and behavioral neuroscience.

References are not in accordance with the journal style

Response: we reorganized the references according to the Healthcare style as provided in zotero.

Reviewer 2 Report

1.     The title unclearly and insufficiently reflects its content. Moreover, should be rephrased with good academic writing

  1. Please note that the abstract section lacks the main aims of this article.

  1. Moreover, please mention in the title and abstract section any phrase to indicate that this article is a review article.

  1. In the abstract section or in any section in whole manuscript, when you mention the word for the first time, please define it as words and not as an abbreviation.

  1. The abstract should have the background, objectives, methods, results, and conclusion

  1. The abstract too short.

  1. Please note that the introduction part in the review article needs at least four comprehensive paragraphs. So, kindly write further paragraphs in the introduction section.

  1. Increase number of references in the introduction part. This is a review article and requires several references (more than 100 references).

9.     Please make double check for the academic writing (needs native speakers in English).

10.  The review article has no method section.  So, I suggest to start directly with section 2.1 Development of the fMRI method

11.  The review article has no result section.  So, I suggest to start directly with section 3.1. fMRI in Tübingen (personal view from 1996-2006)

12.  All figure captions in the result section are too long, and should not be more than 2-3 lines. So, please rephrase it.

  1. Where is the conclusion section? Please write this section.

  1. This is a review article and requires several references. At least 100 references.

  1. Dear authors, I appreciate your efforts to produce this manuscript, yet I would like to make some comments. In my opinion the manuscript still needs some improvements of English.

  1. Abstract - conclusion sections need rewriting in more scientific language.

  1. Make sure that all sentences are linked together

Author Response

Reviewer 2:

  1. The title unclearly and insufficiently reflects its content. Moreover, should be rephrased with good academic writing

Response: We now changed the title to: “Functional MRI in radiology – a personal review”

  1. Please note that the abstract section lacks the main aims of this article.

Response: We now added the following aim here: “We aim to provide some background for (a) historical developments for fMRI, (b) establishment of two research units for fMRI in Radiology in Germany, (c) description of some contributions within selected fields of systems neuroscience, clinical neurology and behavioral psychology.”

  1. Moreover, please mention in the title and abstract section any phrase to indicate that this article is a review article.

Response: We now inserted that in the title and abstract.

  1. In the abstract section or in any section in whole manuscript, when you mention the word for the first time, please define it as words and not as an abbreviation.

Response: We now reviewed the whole text and inserted the terms with out abbreviation before presenting the abbreviated versions.

  1. The abstract should have the background, objectives, methods, results, and conclusion

Response: We are sorry- but we cannot artificially provide these differentiations in a personal review.

  1. The abstract too short.

Response: We now added additional information in the abstract.

  1. Please note that the introduction part in the review article needs at least four comprehensive paragraphs. So, kindly write further paragraphs in the introduction section.

Response: We now added further paragraphs in the Introduction and do now provide 4 paragraphs.

  1. Increase number of references in the introduction part. This is a review article and requires several references (more than 100 references).

Response: We now inserted new references in the Introduction.

  1. Please make double check for the academic writing (needs native speakers in English).

Response: We now asked another scientific neuroscientist with excellent English capacities to correct the text with respect to the English language.

  1. The review article has no method section.  So, I suggest to start directly with section 2.1 Development of the fMRI method

Response: That is very convenient for us- Thank you!

  1. The review article has no result section.  So, I suggest to start directly with section 3.1. fMRI in Tübingen (personal view from 1996-2006)

Response: Again, that is very convenient for us- Thank you!

  1. All figure captions in the result section are too long, and should not be more than 2-3 lines. So, please rephrase it.

Response: We now rephrased the Figure captions according to suggestions.

  1. Where is the conclusion section? Please write this section.

 Response: We now inserted a Conclusion as suggested.

  1. This is a review article and requires several references. At least 100 references.

 Response: We added additional references to fulfill the 100 level.

  1. Dear authors, I appreciate your efforts to produce this manuscript, yet I would like to make some comments. In my opinion the manuscript still needs some improvements of English.

Response: We now asked another scientific neuroscientist with excellent English capacities to correct the text with respect to the English language.

  1. Abstract - conclusion sections need rewriting in more scientific language.

Response: We rewrote these parts now.

  1. Make sure that all sentences are linked together

Response: We deleted unnecessary sentences.

Reviewer 3 Report

The authors shared their experiences of fMRI in two German University Medicine Units, which should be helpful for similar fMRI groups. I have some minor comments:

(1). There is an extra ‘9’ after ‘K.K. Kwong’ (line 69).

(2). The captions of the figure (lines 137-145 for figure 1 and line 236-245 for figure 2) should be centered. I also suggest to delete the text “Figure 1:” and “Figure2:” (in the top), and center the titles.

(3). The link “nytimes.com/2014/06/19/science” doesn’t work (line 267). Please correct it.

Author Response

Reviewer 3:

The authors shared their experiences of fMRI in two German University Medicine Units, which should be helpful for similar fMRI groups. I have some minor comments:

(1). There is an extra ‘9’ after ‘K.K. Kwong’ (line 69).

Response: corrected

(2). The captions of the figure (lines 137-145 for figure 1 and line 236-245 for figure 2) should be centered. I also suggest to delete the text “Figure 1:” and “Figure2:” (in the top), and center the titles.

Response: we centered now titles and capitations

(3). The link “nytimes.com/2014/06/19/science” doesn’t work (line 267). Please correct it.

Response: We now refreshed the actual link:

https://www.nytimes.com/2014/06/19/science/researching-the-brain-of-writers.html?action=click&contentCollection=Science%AEion=Footer&module=MoreInSection&pgtype=article

Round 2

Reviewer 1 Report

Dear authors,

Congratulations on your work!

Reviewer 2 Report

Thanks for your corrections